# Asymmetric cyanation of imines via dipeptide-derived organophosphine dual-reagent catalysis

Hong-Yu Wang[1], Chang-Wu Zheng[1], Zhuo Chai[1], Jia-Xing Zhang[1] & Gang Zhao[1]

Over the past few decades, enantioselective phosphine organocatalysis has evolved rapidly into a highly efficient catalytic strategy for a range of useful reactions. However, as restricted by the traditional catalytic modes, some important reactions, such as asymmetric Strecker-type reactions, have thus far been out of reach of this strategy. Reported herein is an application of enantioselective phosphine organocatalysis for asymmetric Strecker-type reactions, enabled by a dual-reagent catalyst system in which the key organophosphorus zwitterion intermediate, generated in situ by mixing a chiral dipeptide-derived multifunctional organophosphine with methyl acrylate, is used as a highly efficient chiral Lewis base catalyst. The high efficiency of this catalytic system is demonstrated in the asymmetric cyanation of isatin-derived ketimines and azomethine aldimines as well as in the kinetic resolution of racemic 3-substituted azomethines. Mechanistic studies provide experimental evidence for the intermediacy of the putative zwitterion and its role as a catalytically active Lewis base.

[1] Key Laboratory of Synthetic Chemistry of Natural Substances, Shanghai Institute of Organic Chemistry, Chinese Academy of Sciences, 345 Lingling Road, Shanghai 200032, China. Correspondence and requests for materials should be addressed to G.Z. (email: zhaog@sioc.ac.cn).

Enantioselective phosphine organocatalysis has advanced rapidly over the past decades as a powerful tool for the preparation of numerous structurally diverse compounds[1,2]. A general catalytic mode for organophosphine catalysis involves the nucleophilic addition of a chiral organophosphine to an electrophilic reactant (usually an electron-deficient alkene) to form a zwitterion intermediate, which subsequently reacts with an appropriate partner to deliver the desired product and regenerate the phosphine catalyst (Fig. 1a). This traditional catalytic route nicely facilitates control of the stereoselectivity via the formation of a covalent C–P bond between the organophosphine catalyst and the electrophilic reactant. However, this catalytic mode restricts the application scope of the enantioselective organophosphine catalysis to only reactions with an activated alkene/allene reactant, such as the (aza)-Morita-Baylis-Hillman reactions[3,4], Rauhut–Currier reactions[5–8], Michael (or γ) addition reactions[9–17] and related annulations[18–33]. Our research group has been interested in the development and application of chiral amino acid-derived polyfunctional organophosphine catalysis[34]. More recently, we have developed an asymmetric dual-reagent organophosphine catalytic system in which only a catalytic amount of an activated alkene is required to react with a multifunctional phosphine to produce the key zwitterion as an in situ-generated chiral Brønsted base catalyst. This new catalytic

mode dispenses with the need for an activated alkene/allene reactant and thus expands the reaction scope of enantioselective organophosphine catalysis to other types of reactions, such as the Mannich-type reactions (Fig. 1b)[35].

In our continued efforts toward a broader application scope of enantioselective organophosphine catalysis, we envisioned that the key zwitterion intermediate in such a catalytic system might also be a chiral Lewis base catalyst[36,37]. To test this hypothesis, we tested the asymmetric Strecker-type reactions of the nucleophilic additions of trimethylsilyl nucleophiles (Me₃SiNu) to imines, which could provide facile access to chiral non-natural amino acids, as a touchstone. As the activation of Me₃SiCN by a Lewis base is primarily due to the affinity of silicon for the oxygen or fluorine anion, we reasoned that the key zwitterion intermediate in the dual-reagent catalytic system may also be a suitable Lewis base catalyst for the desilylative cyanation reactions (Fig. 1c). Although numerous successful asymmetric catalytic systems of this important type of reaction have recently been developed using various Lewis base catalysts[38,39], such reactions have been out of the reach of enantioselective organophosphine catalysis[40].

Herein, we report a dual-reagent catalytic system consisting of a chiral dipeptide-derived multifunctional organophosphine and methyl acrylate, which serves as a highly efficient Lewis base catalyst for the asymmetric cyanation of isatin-derived ketimines

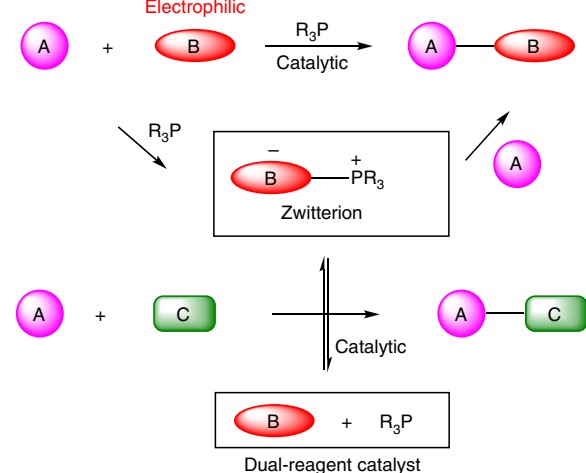

**Figure 1 | Evolution of the organophosphine catalysis to expand the reaction scope. (a)** From general activation mode to dual-reagent catalysis; **(b)** dual-reagent catalyst as a Brønsted base; and **(c)** dual-reagent catalysis as a Lewis base in the cyanation of imines with Me₃SiCN.

and azomethine aldimines with Me₃SiCN as the cyanate source. The versatility of this class of chiral phosphine-based Lewis base catalysts as high-performance catalysts for the asymmetric Strecker-type reaction (at catalyst loadings as low as 0.1 mol%; (ref. 41) is demonstrated.

## Results

**Control experiments with racemic catalysts.** Oxindoles represent an important structural motif that exists in numerous biologically active molecules and natural products. The asymmetric cyanation of ketimines derived from isatins has been an important route to synthetically useful 3-amino-3-cyanooxindoles[42–44]. Initially, we tested the feasibility of the dual-reagent catalytic system using simple achiral organophosphines in the model reaction between ketimine **1a** and Me₃SiCN (Table 1). Various combinations of an organophosphine and methyl acrylate were screened. Notably, the use of a bifunctional organophosphine bearing a thiourea moiety is critical because the reaction barely proceeded when simple triphenylphosphine or methyl diphenylphosphine was used alone or in the presence of methyl acrylate (Table 1, entries 1–3). By contrast, the combination of a bifunctional organophosphine catalyst **B** with methyl acrylate demonstrated the highest catalytic efficiency to provide the desired product in quantitative yield within an extremely short time (<1 min; Table 1, entry 4). In the absence of either the

organophosphine or methyl acrylate, the reactions proceeded much slower (~60% yield after 10 h, Table 1, entries 5–6), whereas a combination of methyl diphenylphosphine, thiourea and methyl acrylate improved the yield to 90% in 10 h (Table 1, entry 7). These results suggest not only the importance of the double H-bond donor in activating the electrophilic imine but also the involvement of a catalytically more efficient species in this combination (for more details, see Supplementary Fig. 1). We also used several aldimines with different protecting groups; to our delight, the reactions all afforded excellent yields (see Supplementary Figs 2 and 3).

**Condition screening.** Encouraged by these results, we next tested different combinations of the designed chiral multifunctional organophosphine-thiourea catalysts with methyl acrylate in the model reaction (Table 2). In general, the proposed dual-reagent catalysis exhibited very high efficiency in all of the examined reactions, providing excellent yields within 10 min at −40 °C even with a very low loading of catalyst (0.1 mol%), although the reaction performed with the organophosphine derived from (R)-1,1′-Bi-2-naphthol [(R)-BINOL] (**3i**) was an exception. With the bifunctional organophosphines derived from simple chiral amino acids (**3a**–**3h**), structural modifications on either the chiral skeleton or the H-bond donor moiety failed to provide satisfactory levels of enantioselectivity. To our delight, when

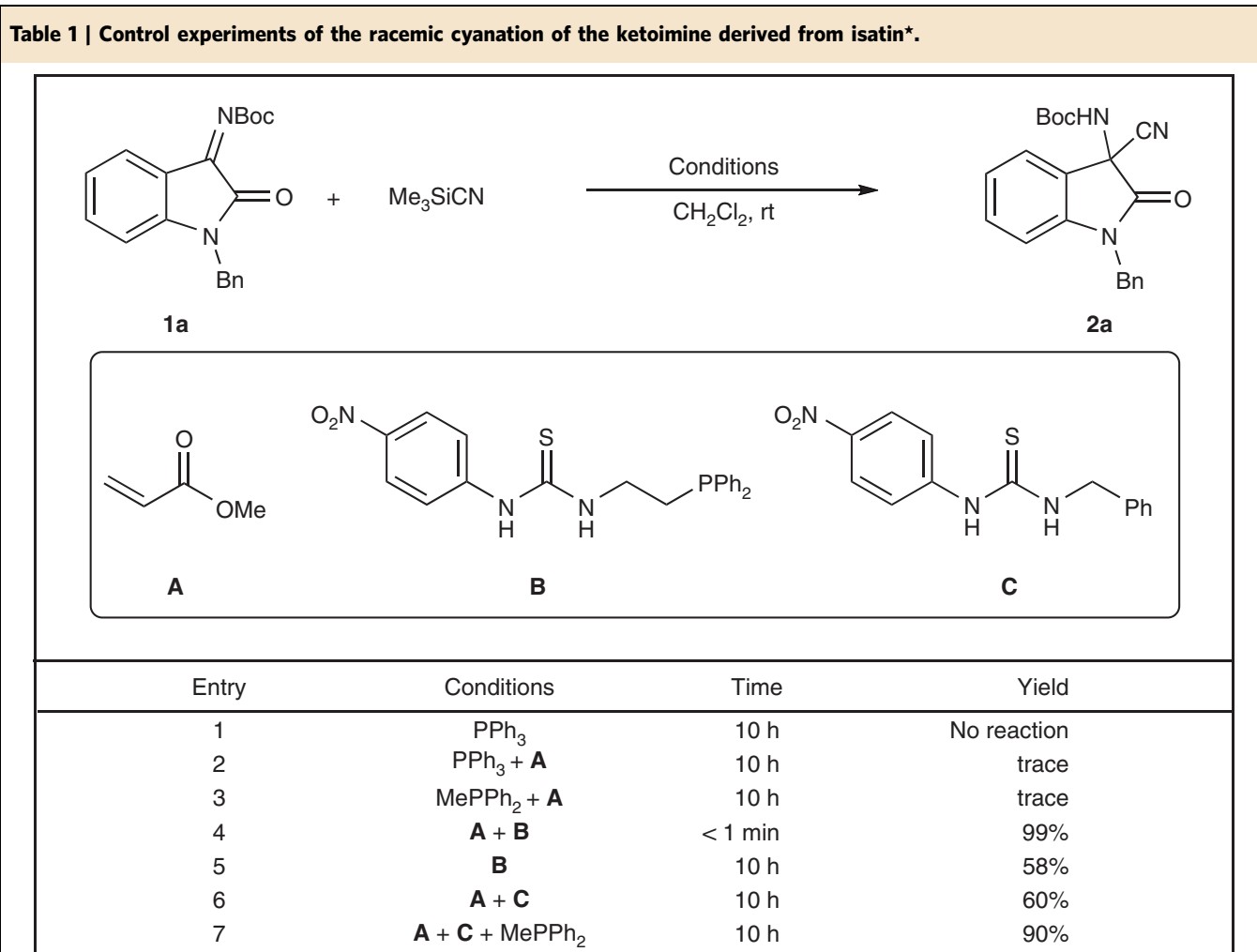

**Table 1 | Control experiments of the racemic cyanation of the ketoimine derived from isatin*.**

| Entry | Conditions | Time | Yield |
|-------|-----------|------|-------|
| 1 | PPh₃ | 10 h | No reaction |
| 2 | PPh₃ + **A** | 10 h | trace |
| 3 | MePPh₂ + **A** | 10 h | trace |
| 4 | **A** + **B** | < 1 min | 99% |
| 5 | **B** | 10 h | 58% |
| 6 | **A** + **C** | 10 h | 60% |
| 7 | **A** + **C** + MePPh₂ | 10 h | 90% |

*Reactions were performed with **1a** (0.12 mmol) and Me₃SiCN (0.2 mmol) in the presence of a catalyst combination (10 mol% for each of the catalyst component if added) for the indicated time. Reported are isolated yields of **2a**.

**Table 2 | Catalyst evaluation\*.**

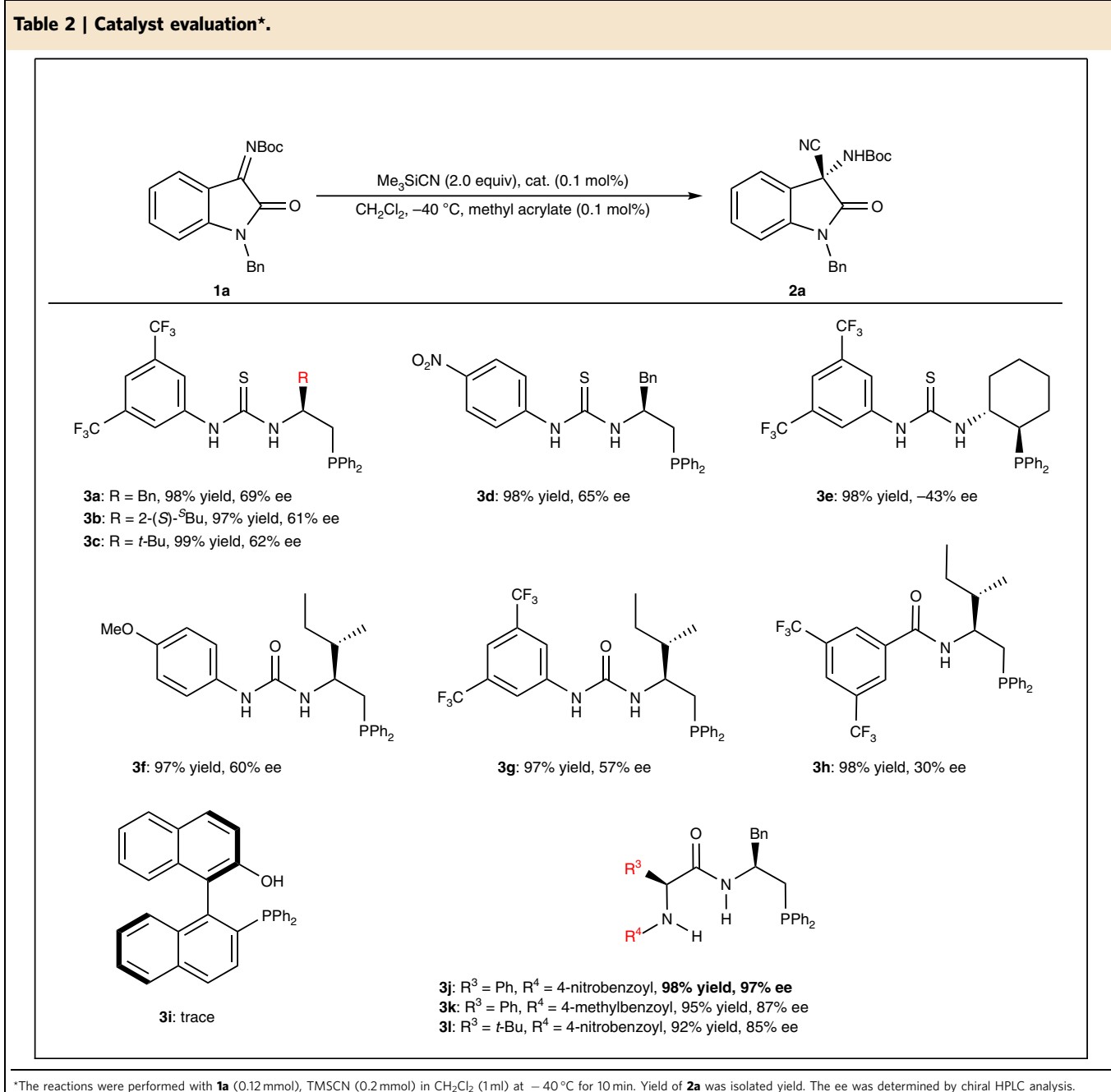

**3a**: R = Bn, 98% yield, 69% ee
**3b**: R = 2-(S)-$^S$Bu, 97% yield, 61% ee
**3c**: R = t-Bu, 99% yield, 62% ee

**3d**: 98% yield, 65% ee

**3e**: 98% yield, −43% ee

**3f**: 97% yield, 60% ee

**3g**: 97% yield, 57% ee

**3h**: 98% yield, 30% ee

**3i**: trace

**3j**: R$^3$ = Ph, R$^4$ = 4-nitrobenzoyl, **98% yield, 97% ee**
**3k**: R$^3$ = Ph, R$^4$ = 4-methylbenzoyl, 95% yield, 87% ee
**3l**: R$^3$ = t-Bu, R$^4$ = 4-nitrobenzoyl, 92% yield, 85% ee

\*The reactions were performed with **1a** (0.12 mmol), TMSCN (0.2 mmol) in CH$_2$Cl$_2$ (1 ml) at − 40 °C for 10 min. Yield of **2a** was isolated yield. The ee was determined by chiral HPLC analysis.

several dipeptide-based bifunctional organophosphines **3j–3l** were used, up to 97% ee was achieved with the catalyst **3j**, which was derived from L-phenylalanine and L-phenylglycine. Other reaction parameters, such as the solvent (CHCl$_3$, 90% ee at − 40 °C; toluene, 84% ee at − 40 °C; CH$_3$CN, 40% ee at − 40 °C) and temperature (0 °C, 91% ee in CH$_2$Cl$_2$, − 20 °C, 94% ee in CH$_2$Cl$_2$), were also investigated; however, no higher ee was attained.

**Cyanation of ketimines**. With the optimized conditions in hand, we next investigated the scope of the asymmetric dual-reagent catalysis in the cyanation of ketimines (Table 3). In general, all the reactions proceeded very smoothly within 1 h to afford the products bearing a chiral tertiary amine in excellent yields (97–99% yield) and enantioselectivities (90–98% ee), irrespective of the electronic and steric nature of the substituents (R$^1$) on the

benzene ring of the isatin skeleton. Different protecting groups (R$^2$) on the nitrogen atom of the isatin, including methyl, benzyl, p-methoxybenzyl and p-nitrobenzyl groups, were all well tolerated in the reactions. Notably, a gram-scale reaction of **1a** and Me$_3$SiCN was also carried out to furnish 1.7 g of the desired product **2a** in 96% yield and 97% ee with an extended reaction time. Notably, the efficiency of the dual-reagent catalytic system demonstrated in this reaction, even with a catalyst loading of 0.1 mol%, was substantially greater than the efficiencies of previously reported catalytic systems for this reaction[45].

**Cyanation of azomethine aldimines**. To further test the potential of the dual-reagent system as a chiral Lewis base catalyst in the cyanation of imines, we subsequently applied it to the asymmetric Strecker reaction between aldimines and Me$_3$SiCN (Supplementary Fig. 2); however, no satisfactory results were

**Table 3 | The scope of the asymmetric cyanation of ketoimines*.**

*Unless otherwise noted, the reactions were carried out with **1** (0.1 mmol), TMSCN (0.2 mmol), methyl acrylate (0.1 mol%) in the presence of the catalyst **3j** (0.1 mol%) in CH$_2$Cl$_2$ (1 ml) at − 40 °C for 1 h. Yields of **2** were isolated yields. The ees were determined by chiral HPLC analysis. The absolute configurations of **2** were determined by comparison of the optical rotation values with literature data.
†Reaction was run with 5 mmol of **1a** and 10 mmol of Me$_3$SiCN at − 45 °C for 10 h.

**Table 4 | The scope of the asymmetric cyanation of azomethine imines*.**

| Entry | R | 5 | Time (min) | Yield (%) | Ee (%) |
|---|---|---|---|---|---|
| 1 | C$_6$H$_5$ | **5a** | 5 | 98 | 91 |
| 2 | 2-FC$_6$H$_4$ | **5b** | 5 | 97 | 91 |
| 3 | 2-MeOC$_6$H$_4$ | **5c** | 15 | 95 | 93 |
| 4 | 2-BrC$_6$H$_4$ | **5d** | 5 | 98 | 93 |
| 5 | 3-ClC$_6$H$_4$ | **5e** | 5 | 97 | 91 |
| 6 | 3-BrC$_6$H$_4$ | **5f** | 5 | 97 | 90 |
| 7 | 4-FC$_6$H$_4$ | **5g** | 5 | 94 | 95 |
| 8 | 4-ClC$_6$H$_4$ | **5h** | 5 | 97 | 93 |
| 9 | 4-BrC$_6$H$_4$ | **5i** | 5 | 96 | 93 |
| 10 | 4-MeOC$_6$H$_4$ | **5j** | 20 | 95 | 93 |
| 11 | 4-MeC$_6$H$_4$ | **5k** | 10 | 98 | 93 |
| 12 | 3-CF$_3$C$_6$H$_4$ | **5l** | 20 | 95 | 91 |
| 13 | 1-naphthyl | **5m** | 10 | 95 | 95 |
| 14 | 2-furanyl | **5n** | 40 | 92 | 95 |
| 15 | cyclohexyl | **5o** | 15 | 91 | 93 |

*The reactions were carried out with **4** (0.1 mmol), TMSCN (0.2 mmol) and methyl acrylate (1 mol%) in the presence of **3m** (1 mol%) in toluene (1 ml) at − 30 °C. The absolute configurations of **5** were determined by comparison of the optical rotation values with literature data. Isolated yields. ees were determined by chiral HPLC analysis.

**Table 5 | The scope of the kinetic resolution of azomethine imines\*.**

| | | |
|---|---|---|
| **7a** 49% yield, 79% ee, >20:1 dr | **7b** 48% yield, 71% ee, >20:1 dr | **7c** 45% yield, 82% ee, >20:1 dr |
| (**R**)–**6a** 48% yield, 80% ee | (**R**)–**6b** 49% yield, 67% ee | (**R**)–**6c** 42% yield, 80% ee |
| s = 34 | s = 11 | s = 50 |

| | |
|---|---|
| **7d** 45% yield, 79% ee, >20:1 dr | **7e** 48% yield, 81% ee, >20:1 dr |
| (**R**)–**6d** 47% yield, 82% ee | (**S**)–**6e** 49% yield, 78% ee |
| s = 65 | s = 23 |

*The reactions were carried out with ( ± ) − **6** (0.2 mmol) and TMSCN (0.12 mmol) in the presence of **3m** (10 mol%) and methyl acrylate (10 mol%) in toluene (1 ml) at − 50 °C for 1 h. Yield of **6** and **7** were isolated yields. The ees were determined by HPLC analysis. The diastereoselective were determined by $^1$H NMR analysis. The absolute configuration of **6** were determined by comparison of the optical rotation values with literature data and the absolute configuration of **7b** was further confirmed by X-ray crystallographic analysis. S-factor = K(fast)/K(slow) = ln[(1-conv) (1 − ee$^6$)]/ ln[(1 − conv)(1 + ee$^6$)].

obtained. After some experimentation to optimize the reaction conditions (see Supplementary Table 1), we identified the combination of organophosphine **3m** and methyl acrylate (1 mol% for each) as the best catalyst for the reaction in toluene at − 30 °C. A range of aldimines derived from aromatic aldehydes, regardless of their electronic and steric nature, worked very well in the reaction to furnish the corresponding products in excellent yields and with high ee values (Table 4, entries 1–14). Notably, the aldimine **4o** derived from an aliphatic aldehyde was also well tolerated in the reaction system, giving the product **5o** in 91% yield and with 93% ee (Table 4, entry 15).

**Kinetic resolution of aldimines.** Encouraged by the impressively high reactivity in the reaction with azomethine aldimines **4** and in light of the broad application of the chiral products as useful building blocks in organic synthesis[46], we next tested the kinetic resolution of a series of racemic azomethine aldimines **6** via cyanation mediated by the previously discussed chiral dual-reagent catalytic system (Table 5). Relevant studies addressing the kinetic resolution of azomethine imines are rare, but include Fu's Cu-catalysed [3 + 2] cycloaddition[47] and Chi's NHC-catalysed [3 + 4] cycloaddition to construct dinitrogen-fused heterocyclic structures[48] and Beauchemin's Brønsted acid-catalysed enantioselective reduction of azomethines[49]. Under reaction conditions similar to those listed in Table 4, we resolved various substituted azomethine imines in high yields with good selectivity factors ($11 \leq S \leq 65$) and obtained the corresponding products with moderate to good enantioselectivities.

**Mechanistic studies.** To elucidate the mechanism of such a dual-reagent catalytic system, we carried several control experiments (Fig. 2) and $^{31}$P NMR spectroscopic analyses of the reaction process (Fig. 3, $^{31}$P NMR and Supplementary Fig. 5). As anticipated, the asymmetric reaction hardly proceeded in the absence of methyl acrylate, which supports our hypothesis that the *in situ*-generated zwitterion is the catalytically active species in the

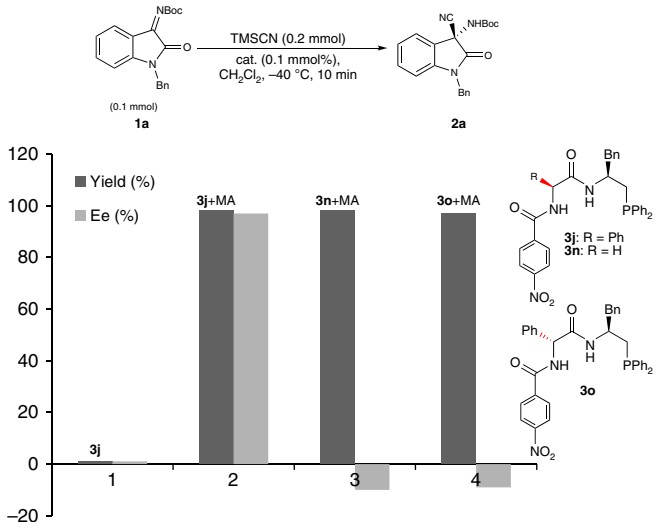

**Figure 2 | Evidence of the asymmetric cyanation of ketimines.** The asymmetric reaction hardly proceeded in the absence of methyl acrylate, and the enantioselectivities obtained with the catalysts **3n** and **3o** with one of the chiral centres removed or reversed were very poor, highlighting the importance of the matched chiral dipeptide skeleton in the enantio-differentiation process.

reaction. The enantioselectivities obtained with catalysts **3n** and **3o** with one of the chiral centres removed or reversed were very poor, highlighting the importance of the matched chiral dipeptide skeleton in the enantio-differentiation process. By contrast, the high modularity of the dipeptide skeleton enabled great flexibility in reactivity tuning and thus has great potential for application in various other related asymmetric reactions (Fig. 2). The $^{31}$P NMR spectroscopy studies on the reaction system provide further support for the assumed key catalytically active species in the reaction. The catalyst **3j** alone showed a $^{31}$P resonance signal at $-24.55$ p.p.m.; the intensity of this signal decreased significantly when **3j** was mixed with methyl acrylate (1:1), and a new resonance appeared at 26.43 p.p.m. This new resonance was assignable to a zwitterion intermediate, which was supported experimentally by the detection of a single peak ($m/z$ for $(M+H)^+ = 688.1$; Supplementary Fig. 4) in the ESI-MS spectrum and was also consistent with observations reported in previous related studies[10,35]. Moreover, when Me₃SiCN was mixed with the above solution of **3j** and methyl acrylate, another new single resonance appeared at $\delta = 30.61$ p.p.m., suggesting efficient activation of Me₃SiCN by the zwitterion intermediate. In sharp contrast, no appreciable change was observed when Me₃SiCN was mixed with the catalyst **3j** alone (Fig. 3, $^{31}$P NMR).

Investigation of the reaction using *in situ* infrared spectroscopy provided further information about the interaction between TMSCN and the catalysts. The band at 2,194 cm$^{-1}$ in the spectra in Fig. 3 was assigned to the stretching of the C≡N bond of TMSCN, and new band at 2,090 cm$^{-1}$ in Fig. 3 was observed when TMSCN and ketimine **3j** were added sequentially. The new infrared band was much clearer in a 3D view, as shown in Fig. 3 (3D IR). The compound TMS–CN (2,194 cm$^{-1}$) has been reported to exist in equilibrium with the isocyanide TMS–NC (2,088 cm$^{-1}$) at room temperature[50]. We propose that the new

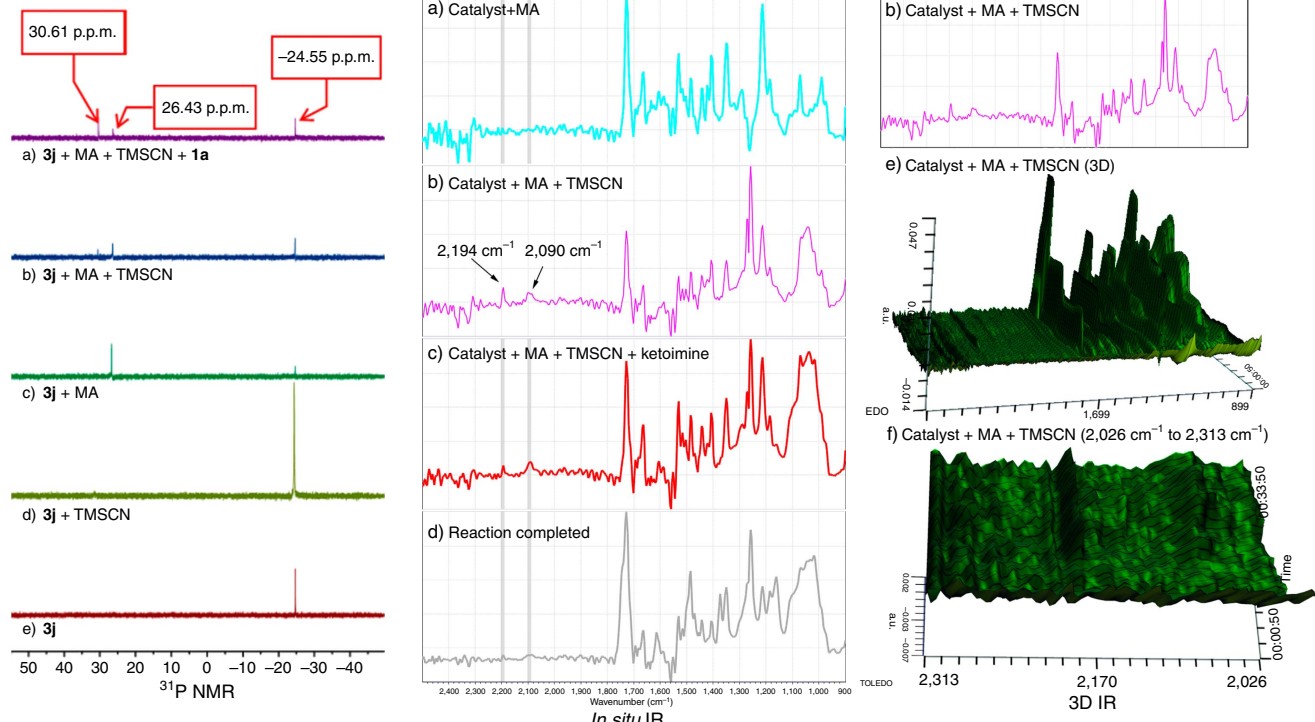

**Figure 3 | $^{31}$P NMR and *in situ* IR studies in CH₂Cl₂.** ($^{31}$P NMR-a) **3j**: MA:Me₃SiCN: **1a** = 1:1:1:1, at the end of the reaction; ($^{31}$P NMR-b) **3j**: MA:Me₃SiCN = 1:1:1; ($^{31}$P NMR-c) **3j**: MA = 1:1; ($^{31}$P NMR-d) **3j**: Me₃SiCN = 1:1; ($^{31}$P NMR-e) **3j** (MA = methyl acrylate); (*in situ* IR-a) **3j**: MA = 1:1; (*in situ* IR-b) **3j**: MA:Me₃SiCN = 1:1:1; (*in situ* IR-c) **3j**: MA:Me₃SiCN:1a = 1:1:1:1; (*in situ* IR-d) At the end of the reaction; (3D IR-e) The integral three-dimensional spectrum of *in situ* IR-b; (3D IR-f) the local-three-dimensional spectrum of *in situ* IR-b.

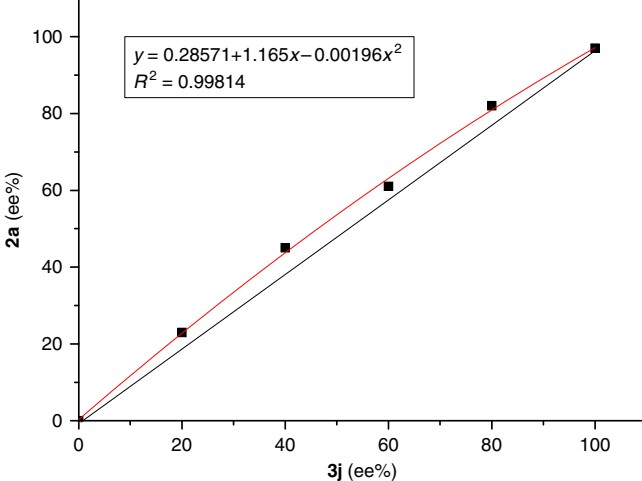

$$y = 0.28571 + 1.165x - 0.00196x^2$$
$$R^2 = 0.99814$$

**Figure 4 | Positive nonlinear effect between ee$_{2a}$ and ee$_{3j}$.** The positive nonlinear effect suggesting the reaction was promoted by an aggregation of the catalyst.

band at $2,090\,cm^{-1}$ is attributable to the more active isocyanide species that isomerized from the cyanide under the catalytic conditions[51]. This may take place through the dimerization of the activated TMSCN with the zwitterion, which is partially verified by the positive nonlinear effect between ee$_{2a}$ and ee$_{3j}$, suggesting the reaction was promoted by aggregation of the catalyst (Fig. 4). The asymmetric amplification in this reaction reflects molecular interactions and complexity in reaction mechanisms. The nonlinearities shown in Fig. 4 may in principle arise by auto association around a matrix of the initial chiral species, through the zwitterion or the dimeric complex of the zwitterion and TMSCN which is an intermediate in the transformation of TMSCN to TMSNC, as demonstrated in the *in situ* IR. Considering the positive nonlinear effect in this reaction is not so obvious and actually the effect is also influenced by other factors such as the concentration, we cannot exclude the possible alternative pathways contributing to the enantioselectivity[52].

On the basis of the aforementioned experimental results and mechanistic studies, we propose a plausible reaction pathway to explain the stereochemical results of the reaction: The mixing of organophosphine and methyl acrylate *in situ* generates the zwitterion intermediate, which then behaves as a Lewis base to promote the isomerization of $Me_3SiCN$ to the more reactive $Me_3SiNC$. This transformation might occur through the dimerization of the activated $Me_3SiCN$ (ref. 51). The active aggregated species then serves as the source of the anionic nucleophile to attack the imine. The hydrogen-bonding interaction between the double-amide N-H and ketoimine (or azomethine imines) enhances the electrophilicity of the imine and drives the nucleophile to approach from the *Re* face (Supplementary Fig. 6).

## Discussion

In summary, we have developed a chiral dipeptide-derived multifunctional organophosphine-based dual-reagent catalytic system and have successfully applied it to the asymmetric cyanation of ketimines derived from isatins. The key finding of this work is that the zwitterion intermediate, which is generated *in situ* by mixing a chiral multifunctional organophosphine with methyl acrylate, could serve as an efficient Lewis base catalyst for asymmetric synthesis. The excellent yields and enantioselectivities (up to 99% yield, up to 99% ee), very low catalyst loading (as low as 0.1 mol%), broad substrate scope, scalability and mild reaction conditions are significant features of the reaction system.

Moreover, we also successfully applied this strategy to the asymmetric cyanation of azomethine aldimines with excellent yields and enantioselectivities as well as the kinetic resolution of racemic 3-substituted azomethine imines under similar cyanation conditions. Experimental evidence in support of the zwitterion as the catalytically active species was also provided. We believe that this mode of asymmetric induction could substantially enrich enantioselective organophosphine catalysis chemistry and open new avenues to the development of relevant Lewis base-catalysed enantioselective organic processes.

## Methods

**General procedure for asymmetric cyanation of ketimines (GP1).** To a vial containing a solution of catalyst **3j** (0.1 mol%) and methyl acrylate (0.1 mol%) in $CH_2Cl_2$ (1 ml) was added TMSCN (0.2 mmol) at $-40\,°C$, followed by the addition of the ketoimine (0.12 mmol). The resultant mixture was stirred at $-40\,°C$ until full conversion of the ketoimine was achieved (monitored by TLC). The mixture was directly purified by column chromatography on silica gel to afford product **2**.

**General procedure for asymmetric cyanation of azomethine imines (GP2).** To a vial containing a solution of catalyst **3m** (1 mol%) and methyl acrylate (1 mol%) in toluene (1 ml) was added TMSCN (0.2 mmol) at $-30\,°C$, followed by the addition of the aldimine **4** (0.12 mmol). The resultant mixture was stirred at $-30\,°C$ until full conversion of the aldimine was achieved (monitored by TLC). The mixture was then directly purified by column chromatography on silica gel to afford product **5**.

**General procedure for asymmetric kinetic resolution (GP3).** To a vial containing a solution of catalyst **3m** (10 mol%) and methyl acrylate (10 mol%) in toluene (1 ml) was added TMSCN (0.12 mmol) at $-50\,°C$, and aldimine **6** (0.2 mmol) was subsequently added to the mixture. The mixture was stirred for 1 h at $-50\,°C$ and then purified directly by column chromatography on silica gel to afford product **7** and recover the unconsumed aldimine **6**.

**Data availability.** The authors declare that the data supporting of the findings of this study are available within the article and Supplementary Information files. For the experimental procedures and spectroscopic and physical data of compounds, see Supplementary Methods. For NMR and HPLC analysis of the compounds in this article, see Supplementary Figs 7–77. The CCDC 1407409 (**7b**) contains the supplementary crystallographic data for this paper (Supplementary Table 2). These data can be obtained free of charge from The Cambridge Crystallographic Data Centre via http://www.ccdc.cam.ac.uk/data_request/cif.

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

## Acknowledgements

Financial support from the National Natural Science Foundation of China (Nos. 21272247, 21572247, 21290184) and XDB 20020100 is gratefully acknowledged.

## Author contributions

H.-Y.W., C.-W.Z. and J.-X.Z. performed the experiments. G.Z. conceptualized and designed the catalytic strategy. The paper was written by H.-Y.W., C.-W.Z., Z.C. and G.Z.; H.-Y.W. and J.-X.Z. prepared the chiral phosphine organocatalysts.

## Additional information

**Competing financial interests:** The authors declare no competing financial interests.

