## [Peer review file · Nature Communications]

Reviewers' comments:

Reviewer #1 (Remarks to the Author):

This is an interesting paper that addresses the explosive field of multifunctional phosphine catalysts and their application to interesting organic reactions. The field is quite crowded with many groups reporting very subtle changes on the basic principles. High selectivities are the norm, as are good yields. Also normative have been speculative transition state models with a minimum of experimental back-up.

This paper is stronger than most since it combines a perhaps unsurprising result (the isatin section) with a more novel result, which involves the asymmetric addition to the azomethine ylid. The unified mechanistic analysis is also intriguing.

The principally interesting mechanistic event is the role of the acrylate ester as a player in transferring the cyanide, through silicon activation. This is quite novel. One can see that this could well have been discovered serendipitously.

The mechanistic discussion could stand to be upgraded. The drawings provide no real insight as to the origin of enantioselectivity. If something is favored, it would be nice to see what might be responsible for a pathway that is disfavored.

The entropic demand of the mechanism is also not well-explored. Are there temperature effects? Could these reactions proceed through multimeric transition states, as is the case for many organocatalytic processes conducted in nonpolar solvents, especially when zwitterionic intermediates are involved?

These things could be addressed in a revised manuscript, and if satisfactorily, then a recommendation of acceptance could be possible.

The SI looks like certain things could be upgraded. For example, why four peaks in the chiral HPLC at the top of page S63?

Reviewer #2 (Remarks to the Author):

Zhao and coworkers report an interesting extension of their previously published dual catalysis concept that combines a phosphinethiourea catalyst with an acrylate conjugate acceptor catalyst to form a catalytically active zwitterion in situ. In the present case, this concept was applied to catalytic enantioselective Strecker reactions with isatin-derived ketimines and the kinetic resolution of azomethine imines. Good selectivities were obtained in most cases. This is undoubtedly nice work that clearly belongs in a good chemistry journal. However, in my opinion, the manuscript does not meet the high bar for publication in Nature Communications for the following reasons:

1. The main concept was reported previously.
2. Asymmetric Strecker reactions have been studied extensively. With regard to ketone-derived imines, those based on isatin are known to be the most reactive and "easiest" substrates. No challenging ketimines were used that would significantly expand the repertoire of known transformations.
3. The present method for the kinetic resolution of azomethine imines does not offer any advantages over known methods.

It should be pointed out that the manuscript contains numerous linguistic and grammatical errors,

starting with the first sentence in each the Abstract and the Introduction. These issues should be addressed prior to resubmission.

Author response to Reviewers' comments:

Reviewer #1 (Remarks to the Author):

This is an interesting paper that addresses the explosive field of multifunctional phosphine catalysts and their application to interesting organic reactions. The field is quite crowded with many groups reporting very subtle changes on the basic principles. High selectivities are the norm, as are good yields. Also normative have been speculative transition state models with a minimum of experimental back-up.

This paper is stronger than most since it combines a perhaps unsurprising result (the isatin section) with a more novel result, which involves the asymmetric addition to the azomethine ylide. The unified mechanistic analysis is also intriguing.

The principally interesting mechanistic event is the role of the acrylate ester as a player in transferring the cyanide, through silicon activation. This is quite novel. One can see that this could well have been discovered serendipitously.

The mechanistic discussion could stand to be upgraded. The drawings provide no real insight as to the origin of enantioselectivity. If something is favored, it would be nice to see what might be responsible for a pathway that is disfavored.

A1: The in-situ FTIR of the reaction and non-linear effect (NLE) between ee2a and ee3j have been performed to upgrade the mechanistic studies. The IR study indicated the TMSCN may isomerize to the more active isocyanides species (Rüchardt, C. et al. *Angew. Chem. Int. Ed.* 1991, 30, 893-901) under the catalytic conditions (Figure 3); while the NLE study suggested the reaction was promoted by an aggregation of the catalyst showing the positive non-linear effect (Figure 4). So the previous proposed single-catalyst transition state was removed. The discussion for this part has been added into the context.

Figure 3|The ^{31}P NMR studies (left) in CH_2Cl_2 with a) $3j : \text{MA} : \text{Me}_3\text{SiCN} : 1a = 1 : 1 : 1 : 1$, at the end of the reaction; b) $3j : \text{MA} : \text{Me}_3\text{SiCN} = 1 : 1 : 1$; c) $3j : \text{MA} = 1 : 1$; d) $3j : \text{Me}_3\text{SiCN} = 1 : 1$; e) $3j$ (MA = methyl acrylate). The In-situ IR studies (center and right) in CH_2Cl_2 with a) $3j : \text{MA} = 1 : 1$; b) $3j : \text{MA} : \text{Me}_3\text{SiCN} = 1 : 1 : 1$; c) $3j : \text{MA} : \text{Me}_3\text{SiCN} : 1a = 1 : 1 : 1 : 1$; d) At the end of the reaction; e) The integral three dimensional spectrum of b; f) The local three dimensional spectrum of b.

Figure 4| Positive non-linear effect between ee_{2a} and ee_{3j} .

The entropic demand of the mechanism is also not well-explored. Are there temperature effects? Could these reactions proceed through multimetric transition states, as is the case for many organocatalytic processes conducted in nonpolar solvents, especially when zwitterionic intermediates are involved?

A2: The temperature effect has been evaluated when we optimized the conditions. As anticipated, higher enantioselectivity was obtained at low temperature (0 °C, 91% ee in CH₂Cl₂, -20 °C, 94% ee in CH₂Cl₂). These data have been added into the context.

As you suggested, we have investigated the relationship between the enantiomeric excess of the product and the enantiomeric excess of the chiral catalyst. The experiments have shown a positive non-linear effect, which indicated the reaction was promoted by an aggregation of the catalyst (Figure 4).

These things could be addressed in a revised manuscript, and if satisfactorily, then a recommendation of acceptance could be possible.

A3: Thank you so much for your suggestions. We have carefully addressed the above questions.

The SI looks like certain things could be upgraded. For example, why four peaks in the chiral HPLC at the top of page S63?

A4: Thank you for your comments. We have improved the Supplementary Information. As you mentioned, the HPLC data of products **2a**, **5f**, and **7a** have been re-collected to remove the impurity peaks.

Reviewer #2 (Remarks to the Author):

Zhao and coworkers report an interesting extension of their previously published dual catalysis concept that combines a phosphinethiourea catalyst with an acrylate conjugate acceptor catalyst to form a catalytically active zwitterion in situ. In the present case, this concept was applied to catalytic enantioselective Strecker reactions with isatin-derived ketimines and the kinetic resolution of azomethine imines. Good selectivities were obtained in most cases. This is undoubtedly nice work that clearly belongs in a good chemistry journal. However, in my opinion, the manuscript does not meet the high bar for publication in Nature Communications for the following reasons:

1. The main concept was reported previously.

A5: Actually, the asymmetric concepts by utilizing the chiral phosphine zwitterion to catalyze a variety of reactions were first put forward by our group. This is a breakthrough for the asymmetric catalysis catalyzed by chiral organophosphine catalysts. Previously, this concept was only applied into the Mannich-type reactions in which the zwitterion acted as a Brønsted base. In this study, we successfully applied this concept into the Strecker-type reactions in which the zwitterion acted as a Lewis base. As shown in the context, the activation mode with the catalyst is totally different from the previous studies we have reported. Herein, we think this study based on our own catalytic system also made a large improvement to the asymmetric organophosphine catalysis.

2. Asymmetric Strecker reactions have been studied extensively. With regard to ketone-derived imines, those based on isatin are known to be the most reactive and "easiest" substrates. No challenging ketimines were used that would significantly expand the repertoire of known transformations.

A6: Asymmetric Strecker reactions have indeed been studied extensively, but this work is the first example that demonstrated the Strecker reaction was catalyzed by the chiral organophosphines via a novel activation mode. The catalytic efficiency is also very excellent compared to the reported examples (low catalyst loading, short reaction time). This is the key point we should address. Actually, we have tried several types of ketimines such as the following. Right now they just show low to moderate enantioselectivities ($\leq 50\%$ ee). It is worthy to note that the reactions with azomethine ketimines afforded the products with about 50% ee. We are still optimizing the conditions and would like to report the results later.

2. The present method for the kinetic resolution of azomethine imines does not offer any advantages over known methods.

A7: We think it is hard to compare this kinetic resolution method to the reported ones as they are the totally different pathways to get the different chiral substituted azomethine imines derivatives (Please see ref 47, ref 48, ref 49). In this study, we have developed the asymmetric resolution of racemic azomethine imines by Strecker-type reactions.

It should be pointed out that the manuscript contains numerous linguistic and grammatical errors, starting with the first sentence in each the Abstract and the Introduction. These issues should be addressed prior to resubmission.

A8: Thank you for your comments. We have carefully revised the manuscript and checked the linguistic and grammatical errors. The Abstract and Introduction are remodeled. The main texts are revised and double-checked.

REVIEWERS' COMMENTS:

Reviewer #1 (Remarks to the Author):

I still like the paper and I think it could be published in Nature Communications. Some issues remain for the consideration of the authors and editors. The authors have tried to be quite diligent in their responses to the reviews and this is appreciated. Some of the additional experiments add depth, but they are not overly conclusive. Chief among these is the (very small) positive nonlinear effect that is observed. The discussion in the text is limited to essentially one sentence. It also does not really fully allow for the possible (perhaps likely) scenario whereby multiple pathways contribute to the observed ee.

A second issue is the very low signal to noise in the in-situ IR studies in the relevant region. Are these conducted at the same concentration as the reaction conditions? It also looks to me like these studies could also support multiple pathways.

I am also not sure why the possible transition states are relegated to the Supplementary pages. These seem central to what most readers will be interested in.

The precision of the writing in the manuscript is still not that high. Perhaps some professional editing will be possible.

I also had the chance to read the other review of the MS. I think it is a bit subjective, and I do not think this work is too incremental.

So, all in all, I think the manuscript discloses interesting and important science. The details could stand to be upgraded for a top journal. Perhaps my comments in this regard are useful to the authors.

Author response to Reviewers' comments:

Reviewer #1 (Remarks to the Author):

I still like the paper and I think it could be published in Nature Communications. Some issues remain for the consideration of the authors and editors. The authors have tried to be quite diligent in their responses to the reviews and this is appreciated. Some of the additional experiments add depth, but they are not overly conclusive. Chief among these is the (very small) positive nonlinear effect that is observed. The discussion in the text is limited to essentially one sentence. It also does not really fully allow for the possible (perhaps likely) scenario whereby multiple pathways contribute to the observed ee.

A briefly discussion of the positive nonlinear effect was add into the manuscript as following: The asymmetric amplification in this reaction reflects molecular interactions and complexity in reaction mechanisms. The nonlinearities shown in Figure 4 may in principle arise by auto association around a matrix of the initial chiral species, through the zwitterion or the dimeric complex of the zwitterion and TMSCN which is an intermediate in the transformation of TMSCN to TMSNC, as demonstrated in the in-situ IR. Considering the positive nonlinear effect in this reaction is not so obvious and actually the effect is also influenced by other factors such as the concentration, we cannot exclude the possible alternative pathways contributing to the enantioselectivity (See: *Angew. Chem. Int. Ed.* **2009**, *48*, 456-494).

A second issue is the very low signal to noise in the in-situ IR studies in the relevant region. Are these conducted at the same concentration as the reaction conditions? It also looks to me like these studies could also support multiple pathways.

The in-situ IR was conducted at the same concentration as the reaction conditions. As the signal to noise is not so strong, the signal in 3-dimensional images was expanded for clarity, as indicated in Figure 3. The IR study suggested the transformation of CN to NC, which may undergo through the dimerization of the TMSCN catalyzed by zwitterion. In this view, the IR study also supports the multiple pathways.

I am also not sure why the possible transition states are relegated to the Supplementary pages. These seem central to what most readers will be interested in.

As the details of the reaction mechanism are still not quite clear (e.g. how the catalyst promoted transformation of CN to NC happened?), we think it is better to put the simple transition states in the Supplementary Information. We are still working to disclose the details of the mechanism especially the interaction between the catalyst and Me₃SiCN. This would be published in a following work.

The precision of the writing in the manuscript is still not that high. Perhaps some professional editing will be possible.

We have used the Nature Publishing Group Language Editing service to improve the writing of this manuscript.

I also had the chance to read the other review of the MS. I think it is a bit subjective, and I do not think this work is too incremental.

Thank you for your supportive comments.

So, all in all, I think the manuscript discloses interesting and important science. The details could stand to be upgraded for a top journal. Perhaps my comments in this regard are useful to the authors.